# Responses of CIPS/AIM Noctilucent Clouds to the Interplanetary Magnetic Field

Liang Zhang[1], Brian Tinsley[2], Limin Zhou[3,4]

[1]State Key Laboratory of Marine Geology, Tongji University, Shanghai, 200092, China
[2]Physics Department, University of Texas at Dallas, Richardson, Texas, 75080, USA
[3]Key Laboratory of Geographic Information Science, East China Normal University, Shanghai, 200062, China
[4]State Key Laboratory of Numerical Modeling for Atmospheric Sciences and Geophysical Fluid Dynamics, Beijing, 100029, China

*Correspondence to*: Liang Zhang (Liangzhang420@tongji.edu.cn)

**Abstract.** This study investigates the link between the interplanetary magnetic field (IMF) $B_y$ component and the Noctilucent clouds (NLCs) measured by the Cloud Imaging and Particle Size (CIPS) experiment onboard the Aeronomy of ICE in the Mesosphere (AIM) satellite. The mean ice particle radius in NLCs is found to be positively/negatively correlated with *IMF* $B_y$ in the Southern/Northern Hemisphere (SH/NH), respectively, on a day-to-day time scale in most of the 20-summer seasons during the 2007-2017 period with a near 0-day lag time, and the response in the SH is stronger than that in the NH. Moreover, the albedo, ice water content, and frequency of occurrence of NLCs present positive correlation with *IMF* $B_y$ in SH but no significant correlation in NH. The superposed epoch analysis (SEA) further indicates the $r_m$ on average changes by about 0.73 nm after *IMF* $B_y$ reversals, which is significant at 90% confidence level in Monte Carlo sensitivity tests. Our results suggest an *IMF* $B_y$-driven pathway: the influence of the solar wind on the polar ionospheric electric potential affects the nucleation processes in NLCs, and consequently the ice particle radius and NLC brightness.

## 1 Introduction

### 1.1 NLCs

The Noctilucent clouds (NLCs), also known as polar mesospheric clouds (PMCs), are the highest and coldest clouds in the terrestrial atmosphere, forming in the high latitude summer mesosphere at ~83 km altitude, where the temperature can drop to ~140 K or lower. The long-term trends in NLCs are thought to be associated with global climate change. NLCs are susceptible to perturbations from lower atmospheric activities such as gravity waves (Gao et al., 2018) and planetary waves (France et al., 2018). NLCs are strongly influenced by both solar and lunar tides, with diurnal and semidiurnal variations observed in the NLC properties (Fiedler & Baumgarten, 2018; Stevens et al., 2017; von Savigny et al., 2017). NLCs also can be affected by solar activities on various time scales, including solar proton events (Bardeen et al., 2016; Winkler et al., 2012), the 27-day solar rotation (Robert et al., 2010; Thomas et al., 2015; Thurairajah et al., 2017), and the 11-year solar

cycle (Dalin et al., 2018; DeLand and Thomas, 2019; Hervig et al., 2019). To distinguish the contribution of solar activity to polar mesospheric clouds from that of climate change, it is important to clarify the mechanisms of the solar influence on NLCs. Based on the observed modest anti-correlation in NLCs with the 27-day and 11-year solar variations, both photodissociation and dynamic origins have been proposed in which the solar ultraviolet irradiance as characterized by the

Lyman alpha ($Ly$-$\alpha$) index is supposed to play a key role by altering the water vapor and temperature in the NLCs region (Dalin et al., 2018; Thomas et al., 2015), while in general the exact mechanism is still unclear. In this paper, the $IMF\ B_y$ rather than the $Ly$-$\alpha$ is applied as the solar activity index to explore the solar wind-NLC link, and a new hypotheses will be discussed in the next section.

## 1.2 $IMF\ B_y$-related mechanisms for NLC-Solar link

The main $IMF\ B_y$-related process is the change of ionospheric potential in polar cap regions, which determines the flow of the regional downward ionosphere-earth current density $J_Z$. The current flow is part of the global atmospheric electric circuit (GEC), with ionospheric potential being ~250 kV positive relative to Earth's surface, maintained by the global thunderstorms and electrified clouds (Slyunyaev et al., 2019; Williams and Mareev, 2014). The Earth experiences a Lorentz electric field applied by the cross product of solar wind magnetic field and velocity, which is mainly northward/southward

for positive/negative (duskward/dawnward) $IMF\ B_y$, and observations have shown that the $IMF\ B_y$-dependent daily-averaged perturbation of ionospheric potential ranges from -30 to 30 kV at high geomagnetic latitudes and is opposite in the SH/NH (Tinsley and Heelis, 1993).

A possible link may exist between the solar wind $B_y$ variations and polar surface meteorology through the ionospheric potential, which has been supported by a variety of observations, in term of polar surface pressure (Lam et al., 2013),

geopotential height (Lam et al., 2014), temperature (Freeman and Lam, 2019; Lam et al., 2018), and below-cloud irradiance (Frederick et al., 2019; Frederick and Tinsley, 2018; Tinsley et al., 2021). It should be noted that these observations are characterized by two features: the responses in SH and NH are opposite, in line with the opposite $IMF\ B_y$-induced ionospheric potential in SH and NH; the delay time is short, lasting only a few days or less. A hypothesis has been proposed to explain the above observations: firstly, solar wind $B_y$ induces changes in the ionospheric electric potential, as well as the

downward current density $J_Z$ in the GEC; second, the microphysical processes inside clouds are sufficiently sensitive to space charge generated by $J_Z$ so that the cloud properties such as infrared opacity and albedo will consequently be affected. Finally, polar surface meteorology will be influenced by cloud radiative forcing (Lam and Tinsley, 2016). The invoked cloud microphysical changes have been detailed for individual aerosol-droplet collisions (Zhang et al., 2018, 2019; Zhang and Tinsley, 2017, 2018), but direct measurements in clouds and modelling are required to test this hypothesis.

In comparison with the tropospheric clouds within which many factors are involved, the polar mesospheric clouds provide a relatively pure scenario to study the role played by electric charges in the microphysical process of clouds. By extending the above 'solar - GEC - cloud microphysics - tropospheric meteorology' hypothesis, it is straightforward to propose the '$IMF\ B_y$ - Ionospheric potential - NLC Microphysics - NLC brightness' hypothesis for the polar mesospheric

clouds: *IMF B_y* induces changes in polar ionospheric potential, which will modulate the charge distribution on meteoric smoke particles (MSPs) with major implications for the nucleation rate and ice particle formation processes in NLCs, and ultimately affect the macroscopic properties of NLCs.

**1.3 Nucleation processes in NLCs**

The formation of ice particles in NLCs is still not well understood, as a variety of factors are involved in the microphysical process, among which the nucleate rate and number density of ice nuclei contribute the most important uncertainties (Rapp and Thomas, 2006). Although the homogeneous nucleation has been considered feasible (Murry and Jensen, 2010), the extreme conditions required make the homogeneous nucleation unlikely to occur at the typical mesospheric supersaturation level (Tanaka et al., 2022). The heterogeneous nucleation instead is thought to be more effective by providing a pre-existing ice nuclei, for which candidates such as ion clusters, soot, sulphate aerosols, meteoric smoke particles have been proposed (Rapp and Thomas, 2006). MSPs are abundant in the mesosphere and considered to be most likely, evidence that ice particles contain small amounts of MSPs has been provided by observations (Hervig et al., 2012). The exact nucleation process of MSPs is still poorly known, due to the lack of laboratory measurements at the mesospheric condition.

The MSPs are generated by meteor ablation at the upper mesosphere and lower thermosphere, with the radius ranging from sub-nanometre to nanometre size. The 2-D simulations involving the middle atmospheric circulation revealed that the MSPs will move upward along with the strong updrafts in the summer mesosphere, and are then transported to winter mesosphere by the meridional winds, and finally sink down into the stratosphere by the downwelling (Megner et al., 2008a, 2008b). The global mass re-distribution of MSPs results in a pronounced reduction of MSPs concentration and lifetime at summer mesosphere, and thus the conventional idea of nucleation on MSPs is challenged.

The above dilemma can be resolved when the charged MSPs are taken into consideration, because the MSPs charge can effectively reduce the critical radius of ice nuclei at low temperature, allowing the charged MSPs to act as ice nuclei (Gumbel and Megner, 2009; Megner and Gumbel, 2009). It should be noted that the galactic cosmic rays can generate continuous ions throughout the atmosphere, and the charged molecular clusters are found to grow much faster than neutral clusters. The so-called ion-mediated nucleation (IMN) is of great important for the formation of cloud condensation nuclei in atmosphere and has been studies for decades (Yu and Turco, 2000; Yu et al., 2008). The distribution of charges on MSPs becomes important with regards to the above assumption, while the efficiency of MSPs collecting electrons at the mesosphere is still unclear. Due to the mobility of electrons is much larger than that of positive ions, negatively charging is supposed to be dominant in the upper mesosphere, and rocket-borne measurements show that about 10% of MSPs are negatively charged (Plane et al., 2014; Robertson et al., 2014). The NLCs locate in the D-region ionosphere where the electric environment is sensitive to disturbances from solar winds. This provides a possible way through which solar activity may impact the NLCs through an electric-related mechanism.

The CIPS/AIM began observing the NLCs in 2007 and 20-summer-season data in SH and NH from 2007 to 2017 are available now. Therefore, we investigated the hypothetical *IMF B_y*-driven solar-NLC link in this study. The paper is

structured as follows: Section 2 provides a brief description of the CIPS data and solar wind data. Section 3 presents the results of NLC correlation with *IMF $B_y$* during the 20 NLC seasons on the day-to-day scale, as well as the superposed epoch analysis for NLCs response to *IMF $B_y$* reversals. Section 4 discusses the results and Section 5 summarizes our main conclusions.

## 2 Data

### 2.1 CIPS/AIM data

The aeronomy of ICE in the Mesosphere (AIM) satellite was launched on 25 April 2007 to a sun-synchronous polar orbit whose local time is mainly midday-midnight at high latitude regions. The Cloud Imaging and Particle Size (CIPS) experiment onboard AIM comprises a panoramic UV nadir imager, consisting of four cameras operating at 265 nm, with a field of view of 120°×80° and a horizontal spatial resolution of 5×5 km. This platform observes the scattered radiance from NLCs, and images the NLCs of ~40°-85° latitude zone for the summer hemisphere ~15 times per day. The CIPS has provided NLC data from the 2007 summer season until now, in terms of ice particle radius, albedo, and ice water content (IWC), and detailed descriptions of the CIPS data products, calibration, retrieval algorithms, and retrieval uncertainties have been published (Carstens et al., 2013; Lumpe et al., 2013). The CIPS level 2 orbit data provide rectangular images of NLC properties for each of the 15 orbit strips per day, in which a single pixel represents a 25 km2 (5×5 km) area anywhere on the globe and a 5800×1000 km strip region is covered, thus the cloud cover as well as the frequency of occurrence (*FO*) of NLCs can be obtained by counting the number of pixels showing them in the images. This study applied the version 5.20 CIPS polar mesospheric cloud level 2 data to investigate the response of NLCs to solar variations during 10 NLC seasons (from 2007 to 2016) in NH and 10 NLC seasons (from 2007/2008 to 2016/2017) in SH.

### 2.2 Solar wind data

The solar wind $B_y$ data in GSM format were downloaded online from the GSFC/ SPDF OMNI Web interface (https://omniweb.gsfc.nasa.gov/form/dx1.html). In the geocentric solar magnetospheric (GSM) coordinate system, the origin locates at the center of the Earth, *X* points towards the sun, *Z* lies in the plane of the *X* and geomagnetic dipole and is perpendicular to *X* (roughly northward), *Y* completes the righthanded coordinate system, stretching toward the dusk. The solar wind structures are fairly complex, varying from 2-sector to 4-sector and sometimes irregularly, therefore, during a 27-day solar rotation period, the *IMF $B_y$* can reverse 2 or 4 or more times, unlike other solar indexes such as *Ly-α* or F10.7 which show regular 27-day period. In order to apply the widely used SEA method, the key days of $B_y$ reversals are listed in Table 1, which have been selected to ensure that during the 5-day period before and after the key day there is no NLCs data missed and that the direction of *IMF $B_y$* is relatively stable. The *IMF $B_y$* changing from positive to negative (from negative to positive) is denoted by *p2n* (*n2p*). Four groups of dates during 2007 and 2017 are listed in Table 1, corresponding to the *n2p*

(28 cases) and *p2n* (29 cases) reversals during NH summer, and the *n2p* (23 cases) and *p2n* (18 cases) reversals during SH summer, respectively.

## 3 Results

### 3.1 Correlation analysis of day-to-day responses of NLCs to *IMF $B_y$*

Figure 1 shows the variations of the daily-averaged solar wind magnetic field and NLC properties during the NLC seasons from 2007 through 2017. The daily-averaged *IMF $B_y$* varies between -5 nT and 5 nT, as shown in Fig. 1(a-b), and the periods of *IMF $B_y$* variations are complex, as noted above. Fig. 1(c-h) show the intensity of NLCs in terms of mean ice particle radius ($r_m$), mean albedo ($Alb_m$), and mean ice water content ($IWC_m$), while Fig. 1(i-j) show the cloud cover of NLCs, as calculated by counting of pixels, and is linearly proportional to *FO*. In order to diminish noise, the NLC data in the latitude bands 65°-85° are used because the NLCs are rarely observed by CIPS below 65° latitude, and an albedo threshold of $5 \times 10^{-6}$ sr$^{-1}$ was applied. The intensity and coverage of NLC peak ~20 days after the solstice, and show strong seasonal variations, with the exception of the mean ice particle radius, $r_m$.

Figure 2 (left) shows the relationship between daily *IMF $B_y$* with the NLC intensity and covers the anomaly in the 65°-85° latitude zone for 2008/2009 season in the SH, with the anomaly obtained by removing the 40-day running mean. The corresponding correlation coefficients are present in the right panel, indicating a clear positive relationship between the NLC brightness and the $B_y$ component of solar wind magnetic fields in the SH. Figure 3 further shows the correlation coefficients of NLC intensity and coverage anomaly in the 65°-85° latitude zone with *IMF $B_y$* for each of the 20 summertime seasons, from 2007 to 2016 in the NH and from 2007/2008 to 2016/2017 in the SH. To remove the seasonal variation, the solar signals are extracted by subtracting the 40-day smoothed CIPS data. To avoid the no-cloud days, only the CIPS data during the period of 10 days before and 50 days after the solstice day are used (Fig. 1). The link between the anomalous mean ice particle radius $r_m$ with *IMF $B_y$* is conspicuous, with positive correlation coefficients in all of the SH summer seasons and negative correlations in most of the NH summer seasons (Fig. 3a). These opposite responses in the SH and NH are consistent with the opposite ionospheric potential changes in SH and NH caused by *IMF $B_y$*. Further, the response was stronger in the SH, with the correlation coefficient being about twice of that in NH. In NLCs, the larger the ice particle size is, the greater the albedo and IWC are, namely, the mean ice particle radius is normally positively correlated with the albedo and IWC (Lumpe et al., 2013), the 20-seasonal CIPS data show a correlation coefficient of ~0.52 between $r_m$ and $Alb_m$ and of ~0.35 for $r_m$ and $IWC_m$. Conversely, the cloud cover of NLCs will also change in pace with the formation and growth process of ice particle radius, and the 20-seasonal CIPS data also show a correlation coefficient of ~0.48 between $r_m$ and *FO*. It is thus reasonable to speculate that the albedo, IWC, and *FO* will respond to *IMF $B_y$* in concert with ice particle radius, and Fig. 3(b-d) show the correlation coefficients between the anomaly of $Alb_m$, $IWC_m$, and *FO* with *IMF $B_y$* are pronounced in SH, but not in NH.

We have also tried to roughly estimated the column number density of ice particles, $N_{ice}$, based on the CIPS data of IWC and ice particle radius $r$. Assuming the mass of ice particle $m_{ice}$ to be $\rho_{ice}4\pi r^3/3$, where $\rho_{ice} = 0.92 g/cm^3$, then the ice particle concentration $N_{ice}$ will be approximately equal to the ice water content divided by the mass of ice particle, $IWC/m_{ice}$. It is of great interest to study the correlation of ice particle concentration with $IMF\ B_y$, since it can reveal the microphysical process during the NLCs responses to solar wind magnetic fields. The results show that the correlation coefficient between ice particle concentration with $IMF\ B_y$ is -0.14±0.06 in SH and 0.09±0.04 in NH, which are surprisingly opposite from that of $r_m$ and $IWC_m$ shown in Fig. 3. In the dry NLC region, ice particles compete for the limited water vapor, resulting in an anticorrelation between the ice particle concentration and ice particle radius, which have been verified by observation and simulation (Hervig et al., 2009; Wilms et al., 2016). Our above results support this anticorrelation again, implying that the solar wind may firstly increase/decrease the nucleate rate and ice particle number density in NLCs, then decrease/increase the ice particle radius.

NLCs are dominantly influenced by the solar tides with the diurnal variation, and the NLCs occurrences are usually more frequent at the local time of morning (Fiedler & Baumgarten, 2018; Stevens et al., 2017). In addition, the NLCs can also be affected by the lunar tides, and the longitudinal variations in NLCs attributed to the non-migrating lunar tides have been found (Liu et al., 2016; von Savigny et al., 2017). To check whether the local time differences between the descending and ascending branches of the AIM satellite will affect the results in Fig. 3, we separate the CIPS data of the descending and ascending branches into two groups. Similarly, in order to check the longitudinal variations, the CIPS data are divided into two groups in term of the longitude ranges of (-180°,0°) and (0°,180°). The correlation coefficients for the above two scenarios have been calculated and listed in Table 2, and the results for all of them are consistent with the results shown in Fig. 3. In summary, the correlations coefficients are found not affected by the local time variations and longitudinal variations in the CIPS data caused by the tide effects, this further proves that our results are robust.

Furthermore, Figure 4 shows the mean correlation coefficients for time lags varying from -7 to 7 days. The error bars illustrate the standard deviation of the mean, which are calculated from the 10 seasonal correlation coefficients and are also given in Fig. 3 at 0-day lag time. A very short delay time was observed (Fig. 4), with the maximum correlations occurring near zero day, implying a microphysical response in NLCs to $IMF\ B_y$ similar to the short delay time that has also been observed in the solar-troposphere studies. In previous studies of the link between $Ly$-$\alpha$ and NLCs, the proposed mechanisms involving photodissociation, heating, or circulation all required longer time. The photodissociation process accounts for a negatively correlation for the $H_2O$ at the mesosphere and the 27-day solar irradiance variations, with a phase lag of about 6-7 days, which can be attributed to the lifetime of $H_2O$ at that altitudes (Shapiro et al., 2012). Satellite observations showed the time lag for the water response to solar 27-day rotation of about 0-3 days and for the temperature response of about 0-8 days, depending on altitudes; and the time lag between NLC properties variations and solar $Ly$-$\alpha$ ranges from 0 to 3 days in the NH and from 6 to 7 days in the SH, depending on instruments and properties (Thomas et al., 2015; Thurairajah et al., 2017). In contrast, the $IMF\ B_y$-related processes of ionospheric potential changes respond quickly to solar wind magnetic field

reversals. In summary, the nearly zero lag-time of NLC properties responding to *IMF $B_y$* variations implies a mechanism of electro-dynamic origin rather than thermal-dynamic origin.

In order to further verify the response of NLCs to solar wind at different latitudes, the approaches in Fig. 3 were repeated for the five latitude bands of 80°-85°, 75°-80°, 70°-75°, 65°-70°, 60°-65°, respectively. The correlation coefficients of the anomaly of NLC properties with *IMF $B_y$* are shown in Figure 5, and the slope of the anomaly of NLC properties to *IMF $B_y$* are given in Figure 6. Fig. 5(a) and 6(a) show that in SH, the correlation and sensitivity of ice particle radius $r_m$ to *IMF $B_y$* are both greater at higher latitudes, in agreement with the ionospheric potential perturbations caused by *IMF $B_y$* changes, while in NH the correlation and sensitivity are just about half of that in SH but still significant in latitude higher than 65°. For the 60°-65° latitude region, the results are not significant, this may because at lower latitudes the *IMF $B_y$*-induced processes are too weak and because the rare NLC occurrences at lower latitudes entail weaker signal/noise.  Fig. 5(b-d) and 6(b-d) show that the responses of the anomaly of $Alb_m$, $IWC_m$, and *FO* to *IMF $B_y$* are noticeable for high latitude in SH, and obvious for $Alb_m$ only at latitudes higher than 75° in NH, but are not obvious for $IWC_m$ and *FO* in NH. Dividing the slope given in Fig. 6 by the yearly averaged NLC properties in 65°-85° latitudes bands, then the relative slope can be obtained: (0.71±0.16)%/nT in SH and (-0.28±0.08)%/nT in NH for $r_m$, (1.36±0.59)%/nT in SH and (-0.52±0.32)%/nT in NH for $Alb_m$, (0.74±0.48)%/nT in SH and (-0.26±0.28)%/nT in NH for $IWC_m$, (2.28±1.73)%/nT in SH and (-0.38±0.60)%/nT in NH for *FO*, and in consideration of the ~5 nT amplitude of *IMF $B_y$* variation during solar wind reversals, the responses of NLC intensity and coverage to *IMF $B_y$* are not negligible. The correlation coefficient of ice particle column number density $N_{ice}$ with *IMF $B_y$* with can also be obtain for different latitudes varying from 85° to 60°: -0.14±0.06, -0.13±0.05, -0.09±0.03, -0.03±0.04, -0.004±0.07 in SH; and 0.06±0.05, 0.09±0.05, 0.12±0.04, 0.04±0.04, 0.01±0.04 in NH. Again, the correlation coefficient of ice particle concentration with solar wind magnetic field is opposite from that of mean ice particle radius and ice water content. However, it should be noted that due to the detection threshold of CIPS instrument for ice particles with radii greater than 10-15 nm, the variation of the invisible smaller ice particles' concentration is unknown.

In addition, other solar wind parameters such as *IMF $B_z$*, $A_p$ index and *Ly-α* irradiance have also been examined by the same processes; however, no correlations were found for them at 0-day lag time. The solar wind magnetic field line has an Archimedes spiral pattern, i.e., *IMF $B_x$* is negatively proportional to *IMF $B_y$* and a correlation coefficient of about -0.67 between them was obtained during the period of 2007 to 2017, thus similar correlations also exist between *IMF $B_x$* and NLC properties, but with the opposite sign. The *IMF $B_z$* corresponds to a dawn-dusk solar wind electric field, and thus can generate a dawn-dusk ionospheric potential drop for both hemispheres, while the sun-synchronous orbit of AIM is designed to be midday-midnight with rare opportunity to pass the dawn-dusk regions, thus the zero correlations observed for NLCs with *IMF $B_z$* are just as expected.

## 3.2 Superposed epochs for NLCs response to *IMF $B_y$* reversals

The superposed epoch analysis is frequently applied in the studies of atmospheric responses to short-term solar variations, in which solar signals are more obvious and easier to be extracted than for decadal or longer-term variations. Although the NLCs only occur in summer, during the 20-season period of CIPS data enough *IMF $B_y$* reversal cases have been accumulated, as listed in Table 1, allowing the SEA method to be used to explore the NLCs responses. In the SEA method, the ice particle radius distribution is denoted by *f(r)*, where the distribution is of the values of *r* over the array of pixels on a given day. The averages of *f(r)* during 3 days before and 3 days after the key day are denoted by $f_{3-pre}$ and $f_{3-aft}$ respectively, then the changes of ice particle radius distribution $\delta f$ during *IMF $B_y$* reversals are given by $\delta f = f_{3-aft} - f_{3-pre}$. The results of $\delta f$ for the *n2p* and *p2n IMF $B_y$* reversals in SH given in Table 1 are illustrated in Figure 7, with an albedo threshold of $5\times10^{-6}$ sr$^{-1}$. The mean ice particle radius $r_m$ can be calculated by integrating the product of radius and its distribution, $r_m = \sum rf(r)$, thus the changes of $r_m$ during *IMF $B_y$* reversals are obtained by $\delta r_m = r_{m,3\_aft} - r_{m,3\_pre} = \sum r\delta f$, and the values of $\delta r_m$ are given in each panel of Fig. 7. For *n2p/p2n IMF $B_y$* reversals, the polar ionospheric electric potential will increase/decrease in the SH, and the $r_m$ increases/decreases by about 0.88/1.07 nm in SH as shown in Fig. 7. Similarly, the results of NH are illustrated in Figure 8, for *n2p/p2n IMF $B_y$* reversals, the polar ionospheric electric potential will decrease/increase in the NH, the $r_m$ decreases/increases by about 0.25/0.71 nm in NH as shown in Fig. 8. Generally, the ice particle average radius changes by about 0.73 nm during *IMF $B_y$* reversals, and the responses in SH is stronger than that in NH. The results in Fig. 7-8 were subject to Monte Carlo sensitive tests, in which the same number of key days in Table 1 were randomly generated and $\delta r_m$ can be calculated by SEA, by repeating this process for one thousand times, the distribution of $\delta r_m$ are obtained, showing the results in Fig. 7-8 are significant at 90% confidence level.

In addition, we also investigate the responses of NLCs to *IMF $B_y$* reversals for different brightness of noctilucent clouds. The NLCs was ranged into five groups by albedo: $5-10\times10^{-6}$ sr$^{-1}$, $10-15\times10^{-6}$ sr$^{-1}$, $15-20\times10^{-6}$ sr$^{-1}$, $20-25\times10^{-6}$ sr$^{-1}$, $25-30\times10^{-6}$ sr$^{-1}$ respectively. It should be noted that the NLCs with albedo less than $5\times10^{-6}$ sr$^{-1}$ are viewed as noise, and the proportion of NLCs with albedo greater than $30\times10^{-6}$ sr$^{-1}$ are negligible. Figure 9 shows that for varying NLCs albedos the particle radius $r_m$ changes during *IMF $B_y$* reversals are consistent to the result in Fig. 7 and 8, verifying that both the dark and the light NLCs are sensitive to *IMF $B_y$* reversals. On the other hand, the NLCs with greater albedo usually have greater mean ice particle radius, thus the results in Fig. 9 also indicate that both the small and large ice particle sensitive to *IMF $B_y$* reversals. In addition, the results in Fig. 9 also support that the responses of NLCs to *IMF $B_y$* is stronger in SH than that in NH.

## 4 Discussion

Our results support the existence of a link between NLCs and solar wind magnetic fields, characterized by the two features of opposite responses in SH and NH in conjunction with a short lag time of 1-day at most, similar to the previously

introduced solar-troposphere link. The '*IMF $B_y$* - ionospheric potential - NLCs microphysics - NLCs brightness' hypothesis can be applied to explain the *IMF $B_y$*-driven solar-NLCs link: *IMF $B_y$* will firstly change the ionospheric potential as well as the downward electric current $J_Z$ at polar regions, subsequently change the fraction of negatively charged MSPs and the nucleation processes in NLCs, finally the ice particle radius, ice particle concentration, IWC, as well as albedo will be affected.

As introduced in the section 1.2, the increase of *IMF $B_y$* will cause the ionospheric potential as well as the ionosphere-earth current density $J_Z$ in the polar cap to increases/decrease in SH/NH. The downward atmospheric current density $J_Z$ is of great interest in the studies of tropospheric clouds, since positive/negative space charges can be induced at the cloud top/bottom boundaries, which has been verified by in-situ observations (Nicoll and Harrison, 2016). As the electric current flows through cloud boundaries, due to the changes of conductivity, gradients of electric field are created, requiring the formation of space charges according to Gauss's Law (Zhou and Tinsley, 2007; 2012). The NLCs locate at the D-region ionosphere, where the ionization and conductivity are caused by solar radiation and thus increase with altitudes. Similarly, net positive space charges will be accumulated in the NLCs region as the downward current $J_Z$ flows through. Moreover, as the ionization varies nearly exponentially with altitudes in the D-region ionosphere, the gradient of electric field is larger at lower altitudes. As a result, the amount of net space charges accumulated in the bottom of NLCs or lower will be larger than in the upper region of NLCs. Given the ionization rate of the D-region ionosphere depends on solar radiation, the effect of *IMF $B_y$* on the ionization rate as well as positive ions concentration should be negligible, thus the net positive space charges are mainly generated by the reduction of electrons.

The MSPs are dominatingly negatively charged because electrons are easier to collect by MSPs as compared to positive ions, consistent with rocket-borne measurements (Plane et al., 2014; Robertson et al, 2014). In consideration of that the net positive space charges induced by the downward current $J_Z$ will reduce the concentration of elections, then a reduction of negatively charged MSPs is also required. And due to the exponentially changes of conductivity, the amount of negatively charged MSPs in the bottom of NLCs or lower will decreases more significantly than that in the upper region of NLCs. Upward vertical winds are dominant in the summer mesosphere, able to carry the MSPs at the bottom of NLCs or lower to pass through the supersaturation region. As mentioned above, the reduction of negatively charged MSPs at lower altitudes are larger than that at higher altitudes, the effect of current $J_Z$ on the nucleation processes of NLCs through the negatively charged MSPs may be further amplified by the upward winds.

As introduced in section 1.3, the critical radius of ice nuclei for the negatively charged MSPs is smaller than that for the neutral MSPs, and will decrease to nearly zero at extreme low temperature. Based on the assumption that the charged MSPs are more efficient than neutral MSPs to form ice nuclei, the concentration of negatively charged MSPs will play an important role on the nucleation rate in NLCs. In addition, studies show that the decrease of nucleation rate will reduce the ice particle concentration, and given the limited amount of water vapor, larger ice particles will be yielded, and brighter NLC will be observed (Wilms et al., 2016).

Our results can be explained in the following pathway: when the *IMF $B_y$* increases, the ionospheric potential and the downward current $J_Z$ will increase in SH, the net positive space charges increase, requiring a reduction in the number density of negatively charged MSPs in the NLCs region. Therefore, the nucleation rate dominated by the negatively charged MSPs will decrease, less ice particles will be formed. Due to the limited of water vapor, the mean particle radius will be larger, and characters such as the albedo, IWC, and cloud occurrence will increase. Conversely, the response of the downward current $J_Z$
to *IMF $B_y$* in the NH is opposite from that of SH, thus the NLCs in NH changes in an opposite way with that of SH.

Polar mesosphere summer echoes (PMSE) are very strong radar echoes scattered by the electron number density irregularities at the polar summer mesopause altitudes of about 75-100 km, and the electron structures are thought to be caused by the neutral air turbulence in combination with the charged ice aerosol particles in the NLCs (Rapp and Lübken, 2004). Note that the NLCs are absent in the winter hemisphere, whereas polar mesosphere winter echoes (PMWE) were still
observed at much lower altitudes of 55- 85 km. PMWE are suggested to be caused by the neutral air turbulence together with the charged MSPs (Strelnikov et al., 2021). A possible link is expected to exist between PMSE/PMWE with *IMF $B_y$* for two reasons: First, the PMSE is sensitive to ice particle radius and concentration, due to the ice particle can affect the diffusion of electrons (Rapp and Lübken, 2004). Our results show that the ice particle radius is sensitive to solar wind, thus it is necessary to check whether this response has further influence on the PMSE. Second, as mentioned in the above microphysical process,
the *IMF $B_y$* is supposed to have a major effect on the charging process of the MSPs, and the latter plays a more direct role in PMSE/PMWE. In brief, to investigate the response of PMSE/PMWE to *IMF $B_y$* will be helpful for understanding the link between solar wind and mesosphere, while the relevant work is beyond the scope of this paper.

In conclusion, our results suggest a new possible way for the link between solar activity and NLCs. The *IMF $B_y$*-related mechanisms are concerned more about the microphysical process of ice nuclei formation, namely, the charging of MSPs and
its influence on nucleation rate. While the *Ly-α* related mechanism focuses more on the photodissociation, heat, and dynamic processes, which will affect IWC on a longer time lag. Unlike the *Ly-α* irradiance which has a regular 27-day period as well as an 11-year period, the *IMF $B_y$* varies in a more complex way, thus its effect on NLCs, as in the correlations, are not just the 27-day period. To better understand the effect of solar activity on NLCs at different lags, periods, and latitudes, the *IMF $B_y$* and *Ly-α* should both be considered in future works.

**5 Conclusion**

The responses of NLCs to solar wind magnetic fields were investigated using the CIPS/AIM data. Our findings suggest that such a solar-NLC link exists. The mean ice particle radius in NLCs was positively/negatively correlated with the *IMF $B_y$* in SH/NH on the day-to-day time scale in the majority of NLC seasons during the period of 2007-2017, with a short lag time of 1 day at most. The correlation and sensitivity of $r_m$ versus *IMF $B_y$* were stronger in the SH, about twice as that in the NH, and
more conspicuous in higher latitudes. Similar responses of albedo, IWC and *FO* in NLCs with *IMF $B_y$* were also noticeable in the SH but not obvious in the NH. The superposed epoch analysis provides further insights into the mean ice particle

radius responses during *n2p* and *p2n IMF B_y* reversals in SH and NH, and results show the $r_m$ averagely changes by about

0.73 nm following *IMF B_y* reversals, which is significant at 90% confidence level in the Monte Carlo sensitivity tests. The

solar-NLC links are interpreted from the perspective of an *IMF B_y*-driven mechanisms: opposite ionospheric electric

potential changes in SH and NH are induced by the *IMF B_y*, which will change the downward current density $J_Z$ flowing

through the NLCs region and thus influence the charging of MSPs. Given the negatively charged MSPs play an important

role on the nucleation processes in NLCs, then the ice particle radius as well as the brightness of NLCs will be affected.

However, it is necessary to further understand the underlying processes of NLCs proposed in above mechanism, and to

implement and verify them in polar mesospheric clouds modelling.

*Data Availability*. The version 5.20 CIPS polar mesospheric cloud level 2 data files are available on: http://lasp/colorado/edu/aim/. The solar wind magnetic field data are available on the GSFC/SPDF OMNI Web interface: https://omniweb.gsfc.nasa.gov/form/dx1.html.

*Author Contributions*. Liang Zhang, Brian Tinsley, and Limin Zhou conceived the idea together. Liang Zhang analyzed the data and drafted the manuscript. Brian Tinsley and Limin Zhou revised the paper and supervised the research.

*Competing Interests*. The authors declare that they have no conflict of interests.

*Acknowledgements*. This work was funded by the National Science Foundation of China (No. 41905059) and the State Key Laboratory of Marine Geology, Tongji University (No. 1350231101/055). We are especially grateful to the entire AIM program for providing us the continuous CIPS data, and we further wish to acknowledge the OMNI group for providing high-quality solar wind data.

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

480

**Table 1.** Dates (year - day of year) of *p2n* and *n2p IMF B_y* reversals during 2007 and 2017 in NH and SH summer.

| *p2n*, NH summer | | | |
|---|---|---|---|
| 2007-159 | 2007-172 | 2007-184 | 2007-199 |
| 2007-227 | 2008-159 | 2008-186 | 2008-212 |
| 2009-171 | 2009-203 | 2010-176 | 2011-172 |
| 2011-198 | 2012-170 | 2012-181 | 2012-211 |
| 2012-225 | 2013-166 | 2013-220 | 2014-168 |
| 2014-184 | 2014-195 | 2014-208 | 2014-222 |
| 2015-158 | 2015-184 | 2015-211 | 2016-174 |
| 2016-201 | | | |
| *n2p*, NH summer | | | |
| 2007-164 | 2007-180 | 2007-192 | 2007-218 |
| 2008-177 | 2009-160 | 2009-194 | 2009-223 |
| 2010-158 | 2010-189 | 2010-220 | 2011-163 |
| 2011-190 | 2011-219 | 2012-163 | 2012-175 |
| 2012-204 | 2012-221 | 2013-180 | 2014-160 |
| 2014-176 | 2014-188 | 2014-212 | 2015-163 |
| 2015-192 | 2015-218 | 2016-162 | 2016-188 |
| *n2p*, SH summer | | | |
| 2007-351 | 2008-12 | 2008-353 | 2008-364 |
| 2009-23 | 2009-355 | 2010-17 | 2010-357 |
| 2011-16 | 2012-25 | 2012-40 | 2012-343 |
| 2013-2 | 2013-11 | 2013-36 | 2013-355 |
| 2014-19 | 2014-346 | 2015-6 | 2015-36 |
| 2015-362 | 2016-17 | 2016-26 | |
| *p2n*, SH summer | | | |
| 2008-31 | 2008-357 | 2009-8 | 2010-5 |
| 2010-30 | 2011-6 | 2011-25 | 2012-6 |
| 2012-33 | 2012-359 | 2013-6 | 2014-11 |
| 2014-39 | 2014-356 | 2015-20 | 2016-11 |
| 2016-20 | 2016-38 | | |

**Table 2.** The correlation coefficients of NLC properties with *IMF $B_y$* under different selections of satellite branches and longitudinal ranges for CIPS data.

| Data selections | $r_m$ (SH) | $r_m$ (NH) | $Alb_m$ (SH) | $Alb_m$ (NH) | $IWC_m$ (SH) | $IWC_m$ (NH) | $FO$ (SH) | $FO$ (NH) |
|---|---|---|---|---|---|---|---|---|
| All | 0.25±0.04 | -0.13±0.04 | 0.16±0.08 | -0.10±0.07 | 0.11±0.08 | -0.05±0.07 | 0.12±0.08 | -0.03±0.07 |
| Ascending | 0.23±0.04 | -0.09±0.04 | 0.14±0.07 | -0.07±0.06 | 0.10±0.07 | -0.05±0.06 | 0.09±0.07 | -0.00±0.07 |
| Descending | 0.19±0.06 | -0.15±0.06 | 0.15±0.08 | -0.10±0.07 | 0.09±0.08 | -0.04±0.07 | 0.13±0.09 | -0.05±0.06 |
| (-180°~0°) | 0.19±0.07 | -0.08±0.04 | 0.15±0.06 | -0.09±0.07 | 0.08±0.07 | -0.05±0.07 | 0.06±0.07 | -0.03±0.05 |
| (0°~180°) | 0.24±0.05 | -0.13±0.04 | 0.12±0.08 | -0.08±0.05 | 0.09±0.09 | -0.03±0.06 | 0.13±0.08 | -0.12±0.06 |

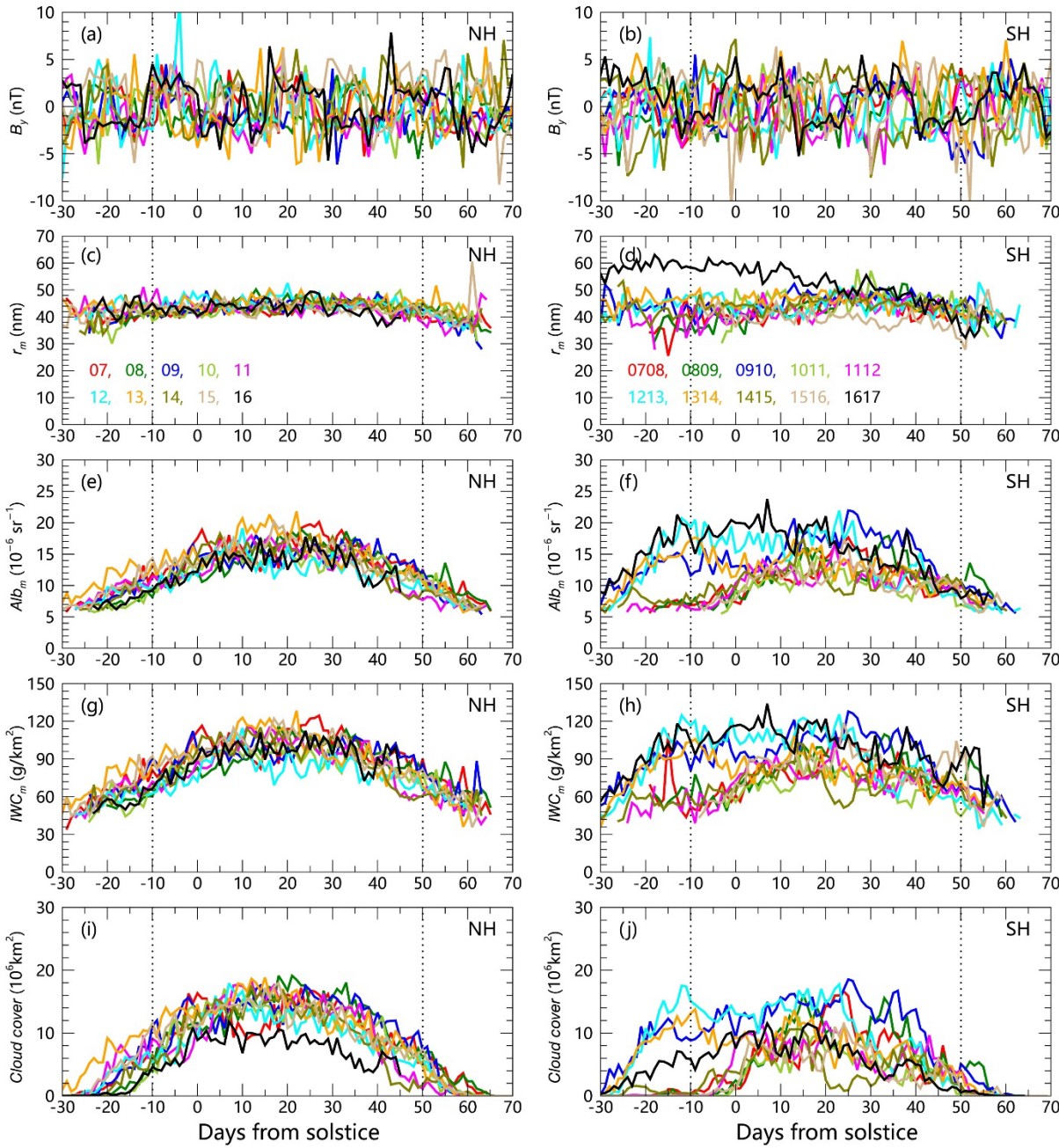

**Figure 1.** Daily-averaged *IMF* $B_y$, mean ice particle radius ($r_m$), mean albedo ($Alb_m$), mean ice water content ($IWC_m$), and cloud cover observed by CIPS for NH (left) and SH (right) for each of the NLC seasons from 2007 through 2017.

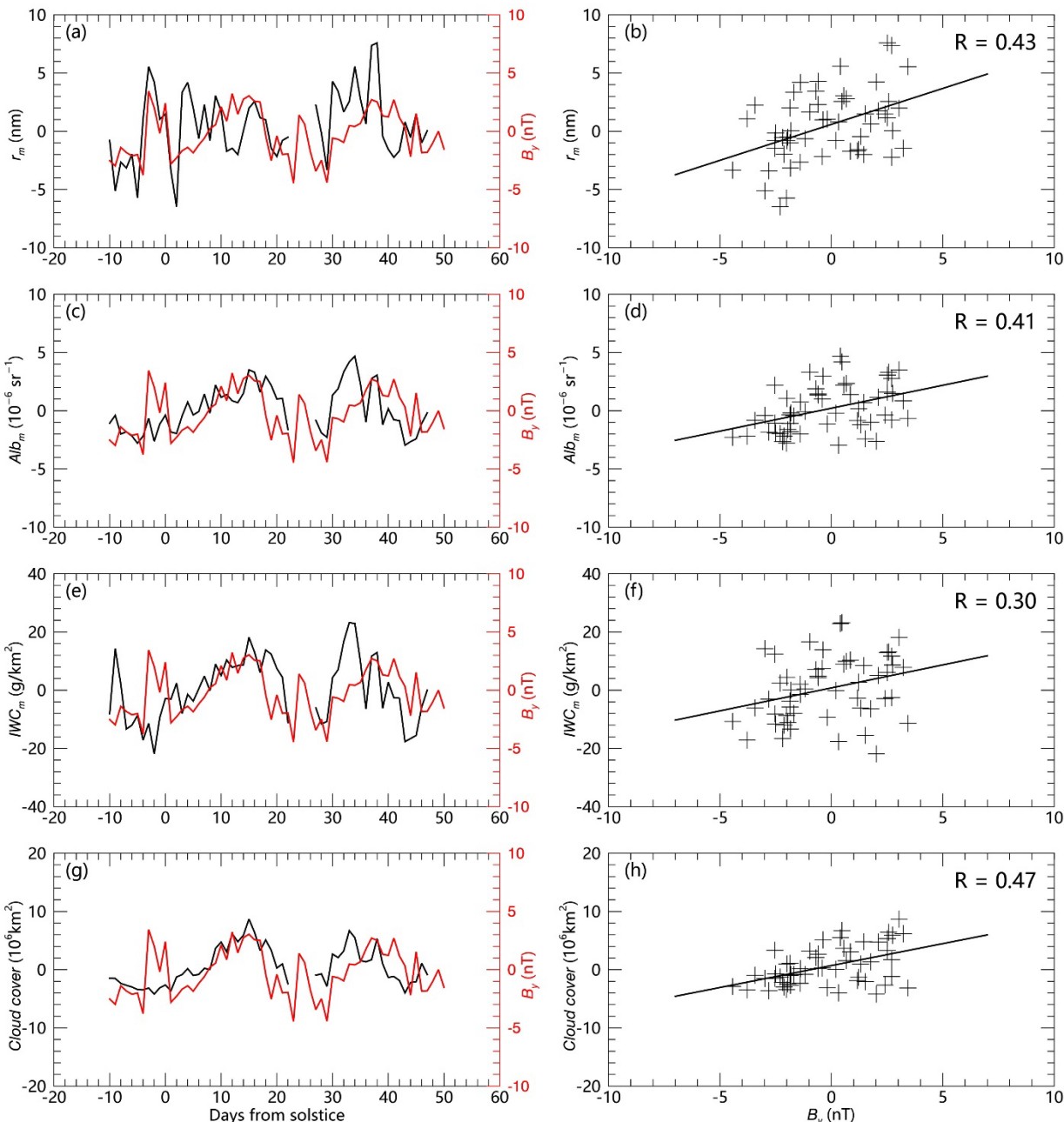

**Figure 2.** The left panels show the relationships of the daily *IMF $B_y$* (red curves) with the anomaly of mean ice particle radius ($r_m$), mean albedo ($Alb_m$), mean ice water content ($IWC_m$), and cloud cover in the 2008/2009 NLCs season for SH. The anomaly of NLCs data are obtained by removing the 40-day running mean. The right panels present the correlation coefficients between the daily *IMF $B_y$* and the anomaly of NLCs characters.

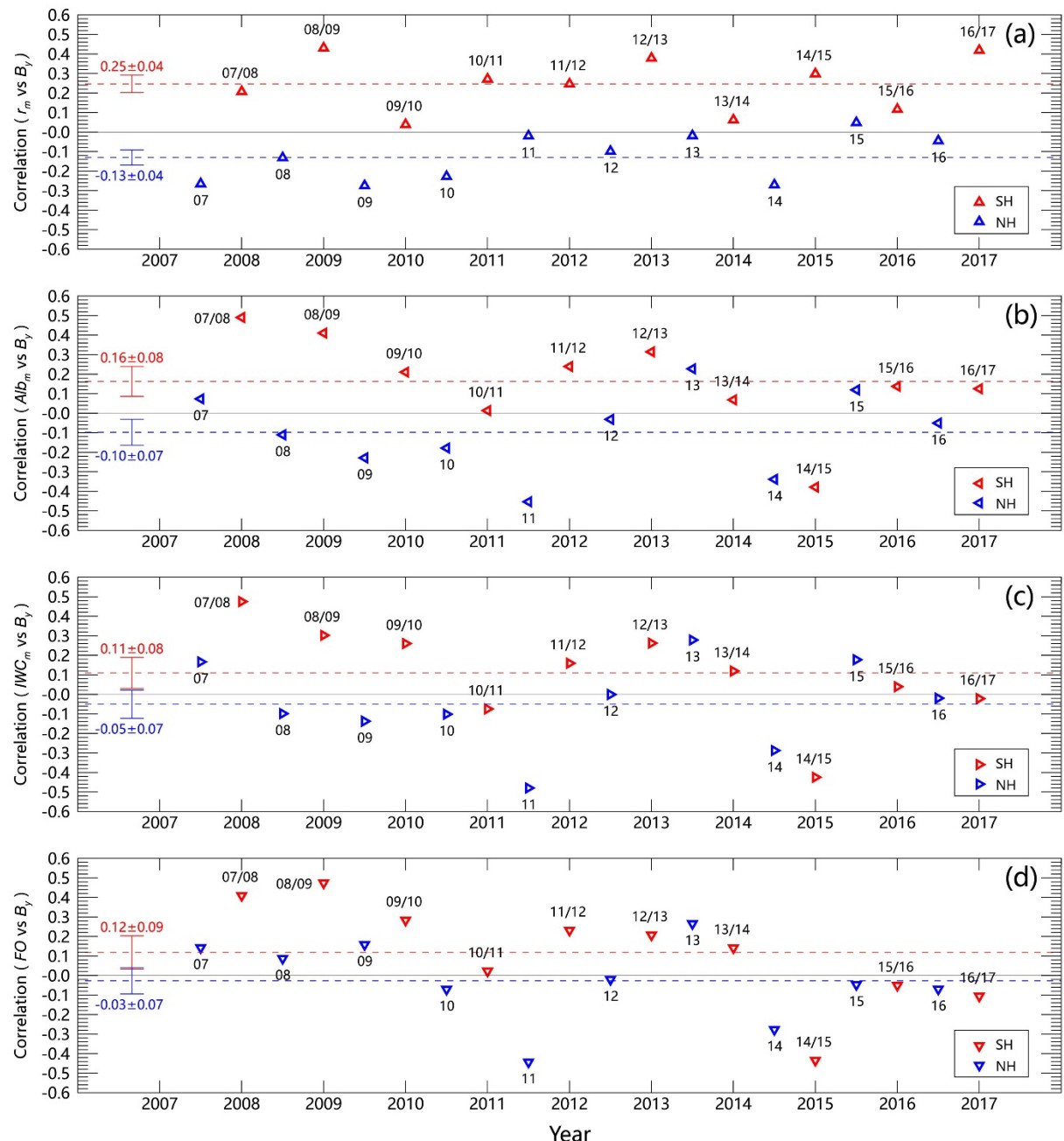

**Figure 3.** Correlation coefficients between the anomaly of $r_m$, $Alb_m$, $IWC_m$ and $IMF\ B_y$ from 2007 to 2017, with red/blue symbols representing the seasonal correlation coefficients and dashed red/blue lines illustrating the mean correlation coefficients for the SH and NH, respectively.

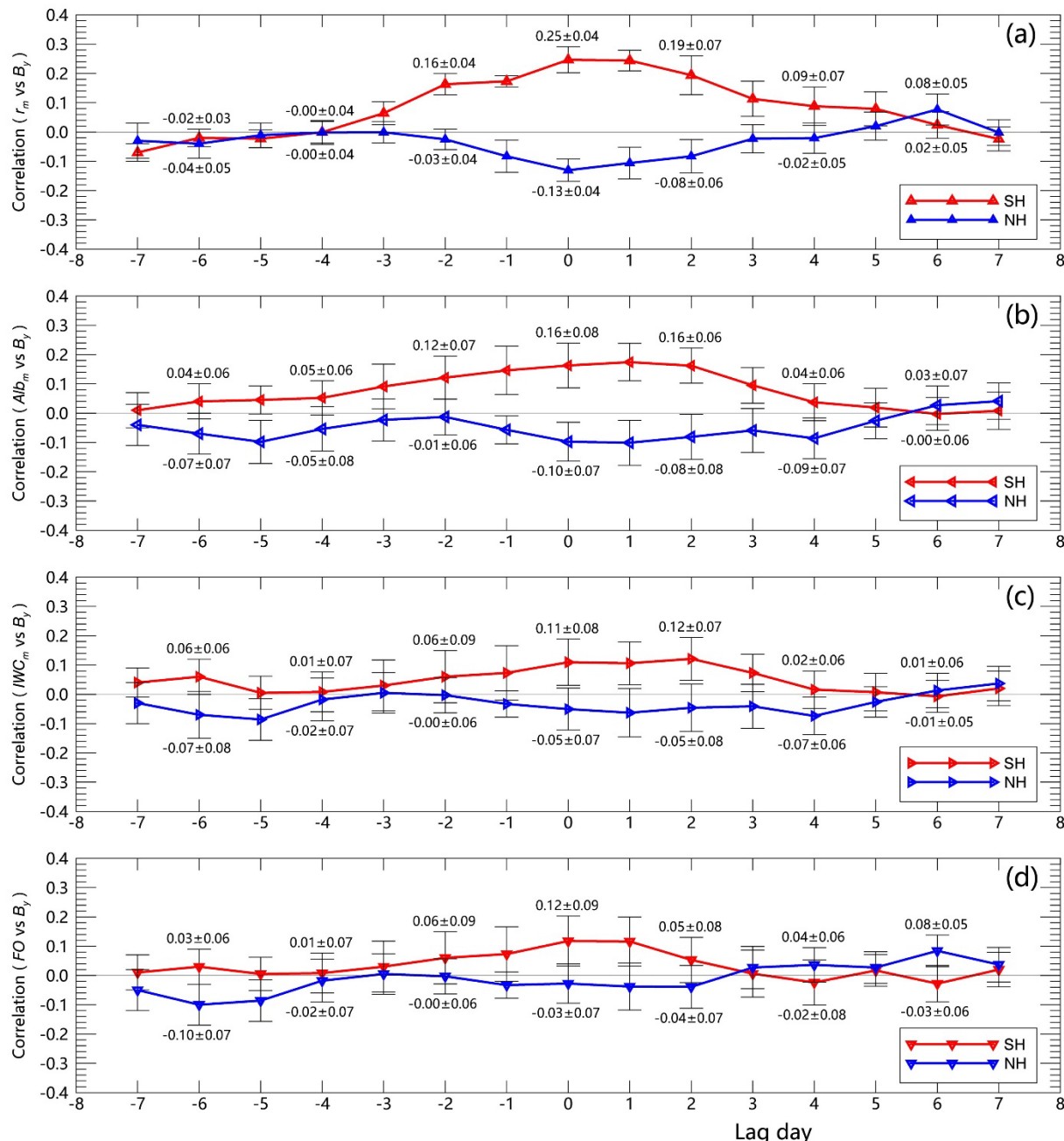

**Figure 4.** Correlation coefficients between the anomaly of $r_m$, $Alb_m$, $IWC_m$ and $IMF$ $B_y$ for time lags varying from -7 to 7 days, with red/blue lines representing the mean correlation coefficients and error bars illustrating the standard deviation of the mean for the SH and NH respectively.

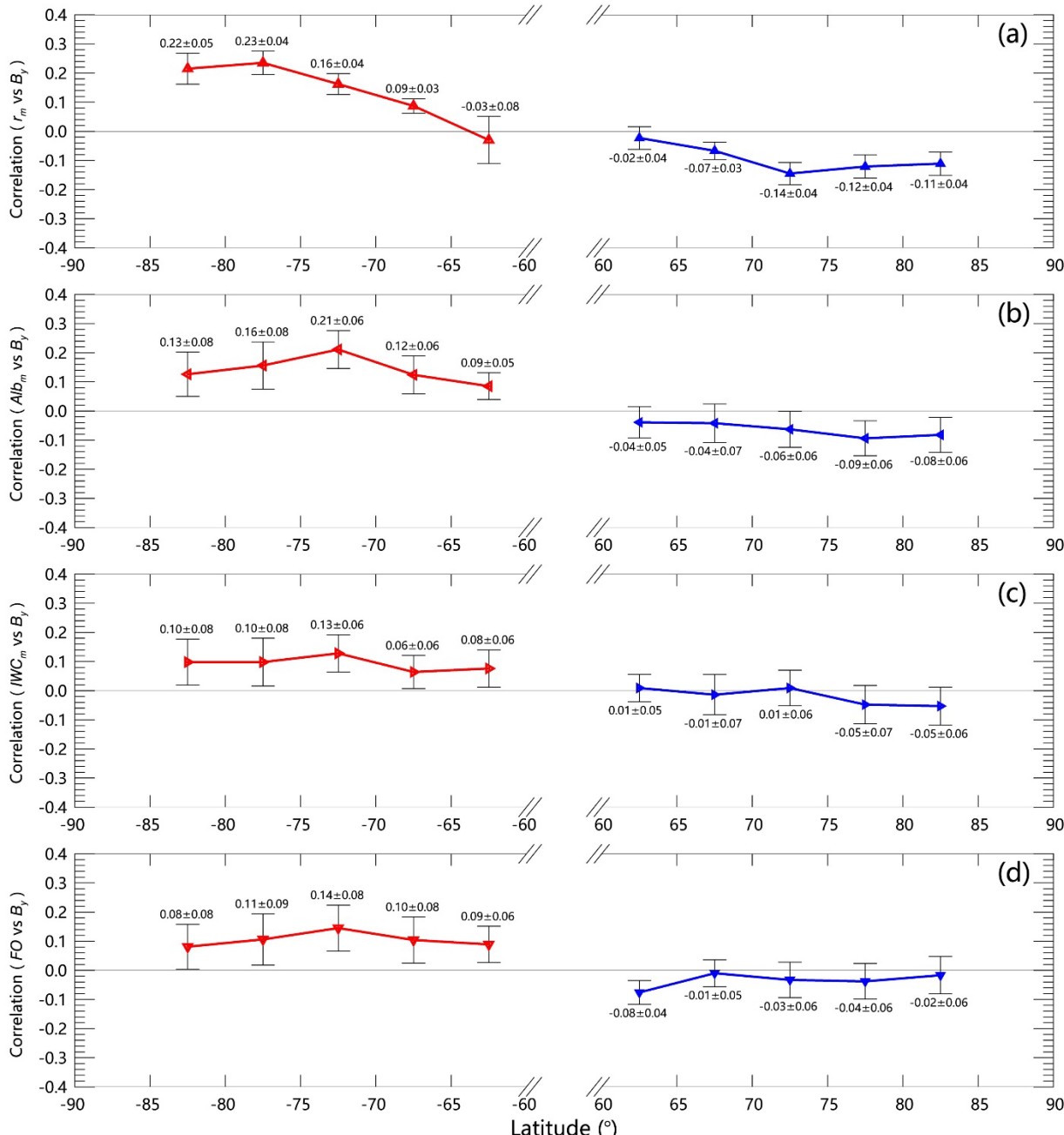

**Figure 5.** Correlation coefficients between the anomaly of $r_m$, $Alb_m$, $IWC_m$ and $IMF$ $B_y$ at different latitude bands, with red/blue lines representing the mean correlation coefficients and error bars illustrating the standard deviation of the mean for the SH and NH respectively.

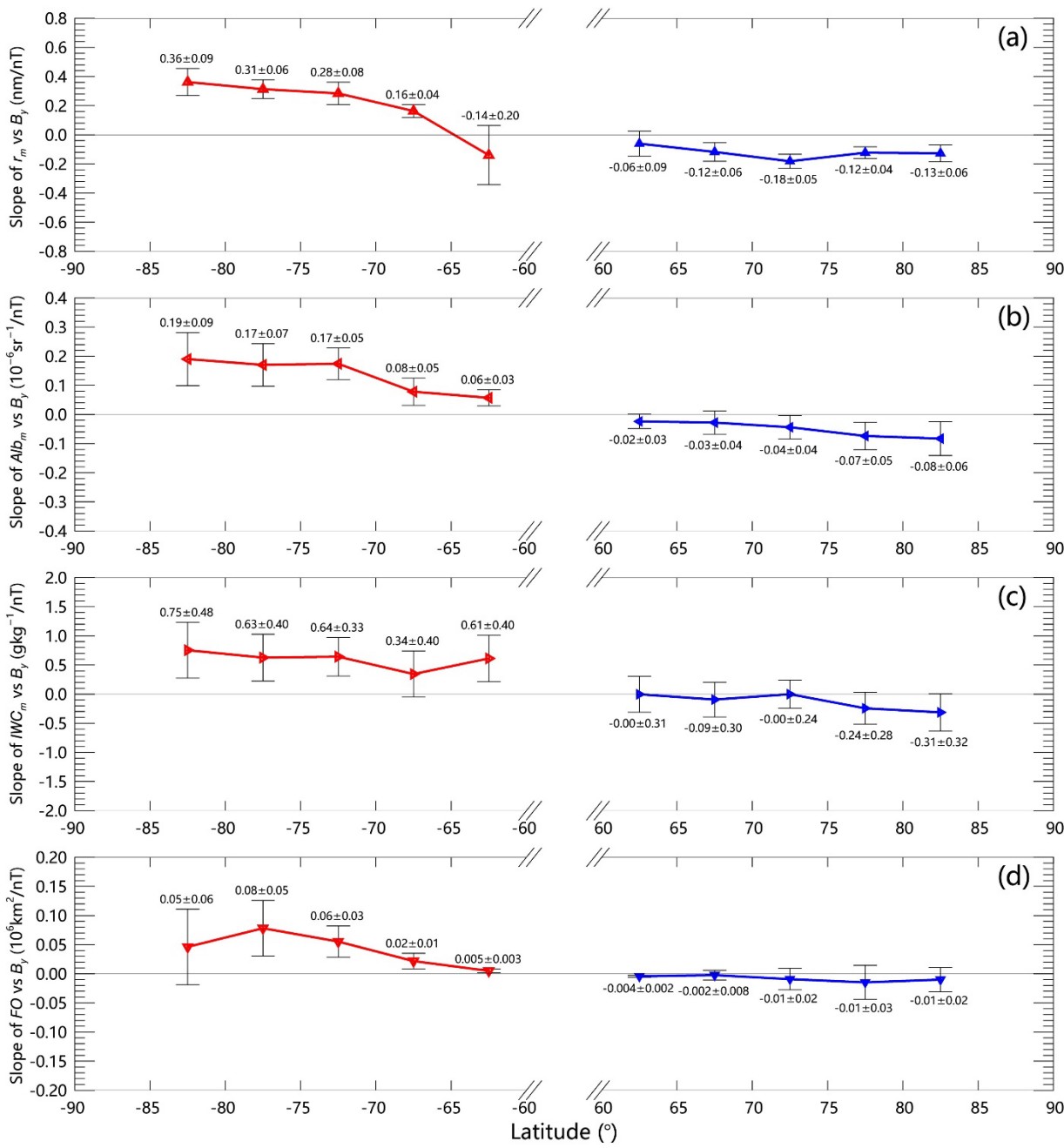

**Figure 6.** Slope of the anomaly of $r_m$, $Alb_m$, $IWC_m$ versus $IMF\ B_y$ at different latitude bands, with red/blue lines representing the mean slope and error bars illustrating the standard deviation of the mean for the SH and NH respectively.

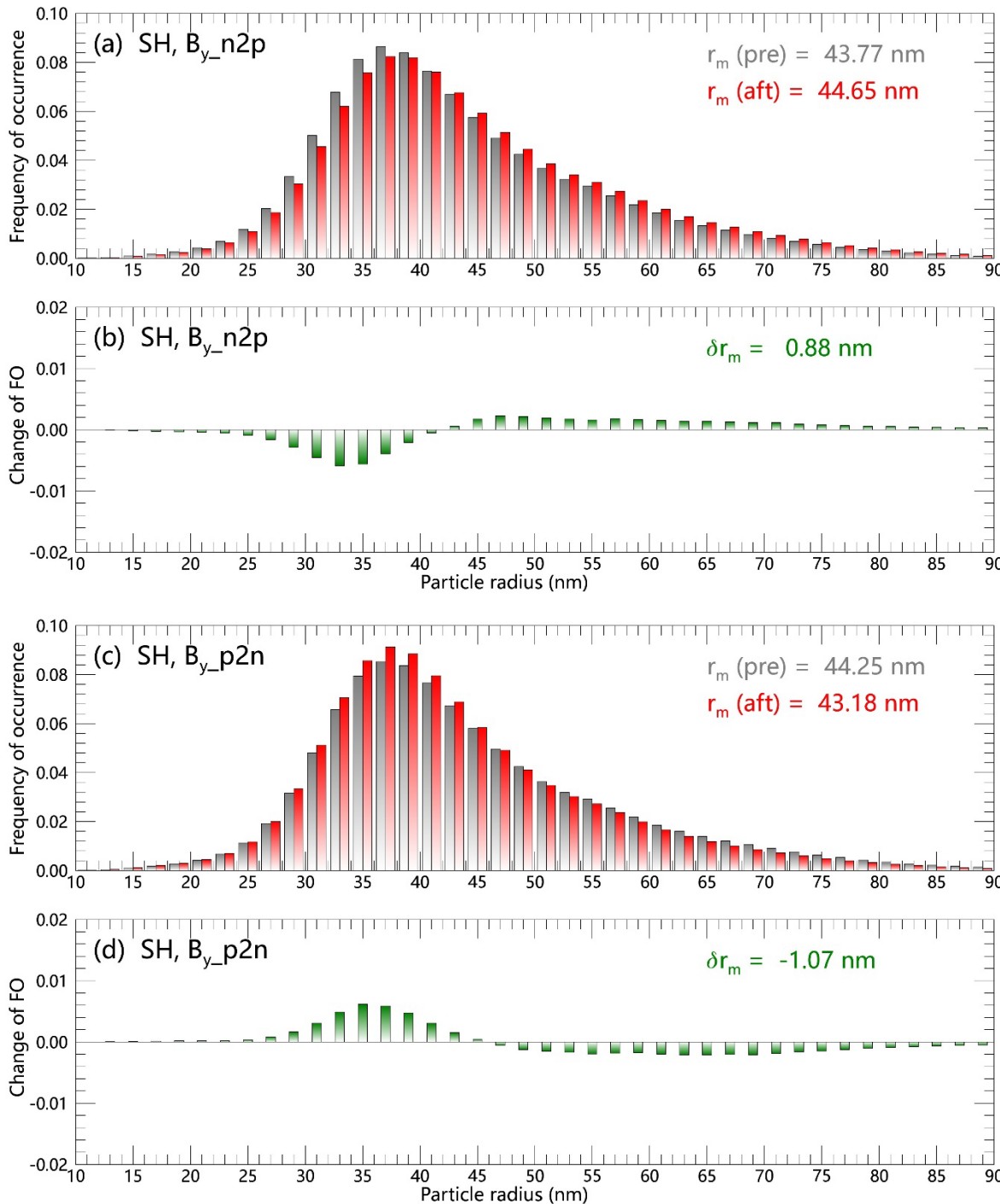

515

**Figure 7.** Changes of ice particle radius distribution *δf(r)* during *n2p* and *p2n IMF $B_y$* reversals in the southern hemisphere. The distributions of *r* over all pixels on three days before/after the key days are indicated by the gray/red bars, and the changes between them are shown by the green bars.

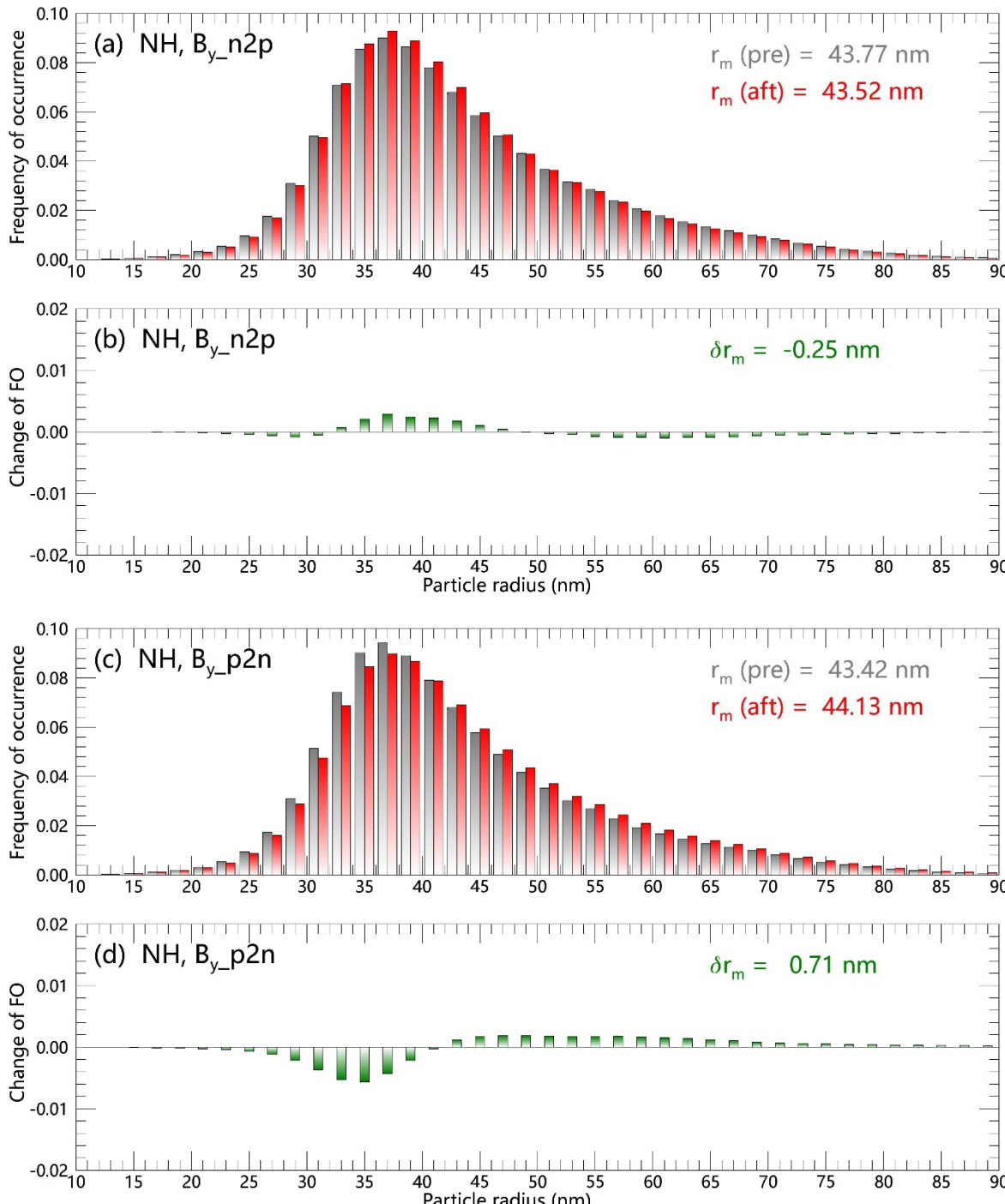

520    **Figure 8.** Similar as Figure 7, but for the results of the northern hemisphere.

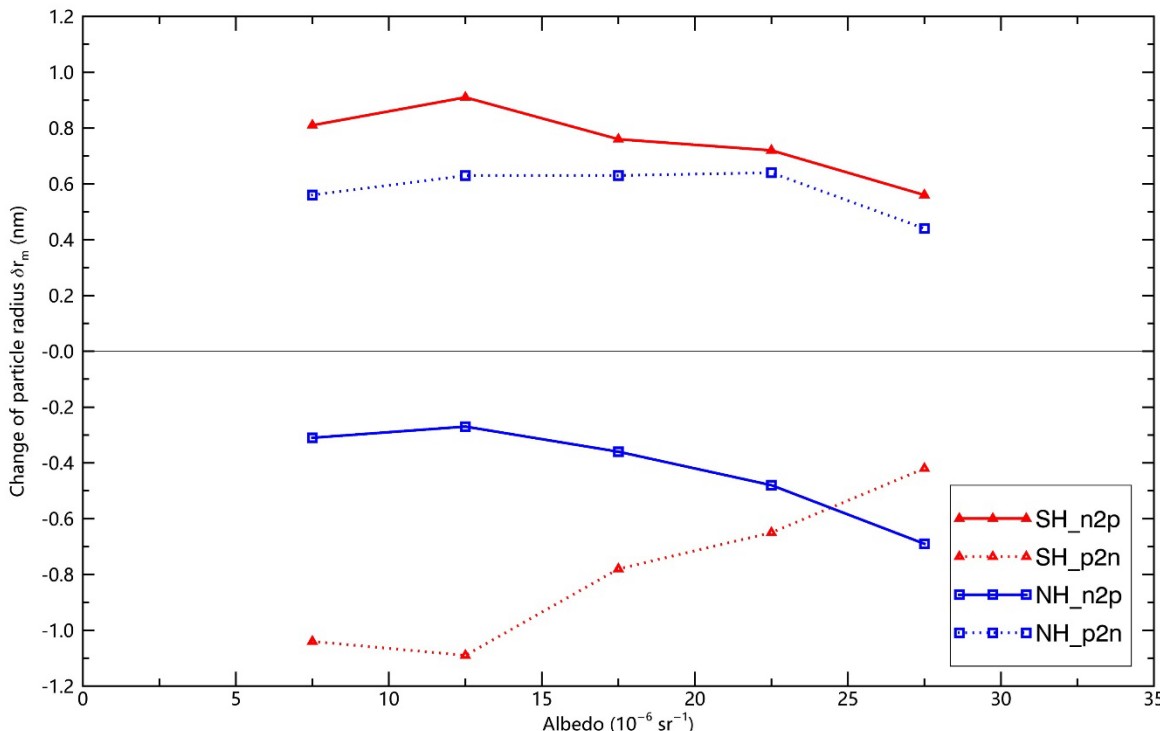

**Figure 9.** The influences of *IMF B_y* reversals on the ice particle radius changes $\delta r_m$ at different NLCs brightness.