# Peer review of "Responses of CIPS/AIM Noctilucent Clouds to the Interplanetary Magnetic Field"

_EGUsphere, 2022_

## Author Comment (AC2)

**Response to Referee Comments on egusphere-2022-126, "Responses of CIPS/AIM Noctilucent Clouds to the Interplanetary Magnetic Field"**

We would like to thank the anonymous referee #2 for his time and effort reviewing our manuscript. We have found the comments to be very insightful, and very helpful to improve our study. Particularly, in line with them a new and more reasonable mechanism has been proposed. All of the following comments have been addressed.

**General Comments**

This paper investigates possible connections between the interplanetary magnetic field (IMF) By component and Noctilucent clouds (NLCs) in Earth's mesosphere. The paper is mostly written well, although there is a tendency for very long sentences, and there are instances when the ideas are poorly expressed.

We thank the referee for the comments. The long sentences have been rewritten to express our ideas more clearly.

The Authors show some reasonably convincing correlations between NLC properties observed by CIPS and measurements of By. Still, the results might be more convincing if there were one or two examples of the By - NLC relationship. For example, they could show a time series of the relevant measurements where we can see that the NLC properties indeed do change concurrent with By variations.

We are very appreciated for this suggestion, and a new Figure has been plotted, which clearly shows a positive relationship of NLCs with *IMF B_y* for the 2008/2009 summer season in the Southern Hemisphere (SH). The time series of them also support that the response of NLCs to solar activity is concurrent with a nearly zero-day delay.

[Figure]

**Figure 2.** The (left) show the relationships of the daily *IMF B_y* with the anomaly of mean ice particle radius ($r_m$), mean albedo ($Alb_m$), mean ice water content ($IWC_m$), and cloud cover in the 2008/2009 NLCs season for SH, the anomaly of NLCs data are obtained by removing the 40-day running mean. The (right) present the correlation coefficients between the daily *IMF B_y* and the anomaly of NLCs characters.

The main problem with this study is that the Authors do not present a believable mechanism that would explain the connection between IMF By and NLCs. They very casually invoke cloud microphysics as a possible explanation, but do almost nothing to explore a plausible pathway. Regarding the microphysics of NLC/PMCs, there are many published studies that could offer some clues here. First off, are the candidates for ice nucleation, which include sulfate droplets, proton hydrates, and meteoric smoke particles (Rapp and Thomas, 2006; Duft et al., 2016), in addition to homogeneous nucleation (Murray and Jensen, 2009). More recent studies indicate that meteoric smoke is contained within NLC particles (Havnes and Næsheim 2007; Hervig et al., 2012), making it perhaps the most likely candidate. Note also that ice - ice coagulation is generally considered unimportant in NLCs. It is relevant that model studies show that increasing the number of ice nuclei can reduce the size of ice particles in PMCs (Megner, 2010), and that changing the ice nucleation rate can alter the concentration and size of NLC particles (Wilms et al., 2016). These later papers may be of particular interest to the present study, and there are certainly more papers to consider than are listed here. The present study would be much more convincing if the Authors present a survey of the relevant literature, and derive a convincing pathway by which the IMF can impact NLC.

These suggestions are very constructive for us, and we are grateful for the referee. We have added a new section in the Introduction part to help understand the results, which includes the nucleate process in NLCs by citing the suggested and relevant references. Moreover, the Discussion part was also re-written to propose a new mechanism, and the key points in the mechamisn are listed here:

1) The basic assumption is that the negatively charged meteoric smoke particles (MSPs) are more effective than neutral MSPs, and changes of the amount of charged MSPs might have a major influence on the nucleate process in NLCs.

2) The NLCs locate in the D-region ionosphere, where the ionization rate caused by solar radiation changes exponentially with altitudes, and thus there will exist significant gradient of conductivity $\sigma$ in this region. According to the Gauss's law, net space charges will be created by the electric field gradient. As a result, when a downward electric current $J_z$ flow through the mesosphere, $E = J_z/\sigma$, net positive space charges will be generated, requiring a reduction in the amount of negatively charged MSPs.

3) The conductivity of the D-region ionosphere varies exponentially with altitudes, the gradient of electric field is larger at lower altitudes, resulting in that the number density of net space charges at the bottom of NLCs or lower is larger than that at the upper region of NLCs. In consideration of the dominate upward vertical winds in the summer pole of mesosphere, the larger changes of negatively charged MSPs concentration at lower altitudes can be transported to NLCs. In short, the upward winds may further amplify the change of charged MSPs number density in NLCs.

4) The nucleation rate will vary in pace with the change of charged MSPs concentration, in that the charged MSPs are quite efficient in forming ice nuclei at low temperature. If the nucleation rate increases, the number density of ice particle will increase, while due to the limited water vapor in NLCs, the radius of ice particle will decrease instead.

5) In conclusion, when $IMF\ B_y$ increases, the ionospheric potential and the downward current $J_z$ will increase in the SH, requiring an increment of net positive space charges

in the D-region ionosphere, thus the amount of negatively charged MSPs will decreased. Therefore, the nucleation process in NLCs caused by negatively charged MSPs will slow down, and the number density of ice particle will also reduce. Finally, due to the competition of the limited water vapor in NLCs, the ice particle radius will increase in the SH. Conversely, the ionospheric potential and the downward current $J_z$ in the NH will decrease when $IMF\ B_y$ increases, thus the ice particle radius will decrease in NH, opposite from that in SH.

This new mechanism is consistent with the results of our paper, and we believe it is more convincing to explain the link between solar activity and NLCs.

It is applicable to this study that the CIPS particle size and IWC results can be used to calculate the column number density of ice particles (i.e., the # of ice particles in the vertical column, #/cm2). This quantity may prove enlightening, especially if you are considering microphysical processes. For example, if ice nucleation is suspect, then the concentration of ice crystals may be expected to change.

This suggestion is quite helpful, we are really appreciated and have applied already. The column number density of ice particle $N_{ice}$ has been estimated by dividing the $IWC$ by the mass of ice particle, that is $N_{ice} \approx IWC\ /\ m_{ice}$, where $m_{ice} = \frac{4}{3}\pi r^3 \rho_{ice}$. The correlation coefficients between ice particle concentration and $IMF\ B_y$ are -0.14±0.06 in the SH and 0.09±0.04 in the NH, which is interestingly opposite from that of the ice particle radius (0.25±0.04 in SH and -0.13±0.04 in NH, as shown in Fig. 2). The above results can be explained by the competition for the limited water vapor in NLCs, where the ice particle concentration and ice particle radius are usually anti-correlated, and therefore their responses to solar activity are supposed to be opposite.

**Specific Comments**

line 23: Here you should introduce the term polar mesospheric cloud (PMC), and state that PMC and NLC are essentially the same phenomena. In the rest of the paper it would be preferred to use only one term, NLC or PMC, but not both.

Done.

line 24: You could state "140K or lower", temperatures of <120K have been observed.

Done, thanks.

line 24: The sentence starting "The long-term trends…" is long and could be 2 sentences.

Done.

line 33: It is not the water vapor and temperature of NLCs, but rather the water vapor and temperature in the NLC region.

Corrected.

line 77: Define the acronym IWC

Done.

line 95: Start a new sentence at the semicolon.

Done.

lines 116-118:   Is there a reference that supports this claim?   Alternately can you include a figure (perhaps a scatter plot) that demonstrates these relationships?

Done. We have added the reference of Lumpe et al., 2013 to support the relationships.

figure 6: The axis label should be frequency of occurrence

Corrected.

line 174: This sentence is confusing. In particular the phrase "by setting the albedo of NLCs varying by 5×10-6 sr-1," is not clear.

Done. We have rewritten this paragraph to make the Figure 8 more readable.

line 186: This sentence continues to line 194, and is far too long. In addition, the ideas here are not expressed clearly. line 192: This statement is unclear. For example, by "the growth of coagulation" do you mean "growth by coagulation"? The next idea, that ice particle coagulation would enhance the formation of ice nuclei, is nonsense. Ice nuclei in the upper mesosphere are likely meteoric smoke particles (there are recent references that discuss this that you should include). Perhaps if ice particle charge had the opposite polarity as smoke particles, then there would be an attraction. In any case. the ideas here are potentially important and need to be more clearly expressed.

Agreed. The above ideas have been abandoned. We have rewritten the Discussion section, focusing on the nucleation process of the charged MSPs, and the new mechanism has been mentioned in details in previous reply.

line 207: Note that Lyman-alpha radiation also varies on an 11-year cycle.

Done.

---

## Author Comment (AC3)

**Response to Referee Comments on egusphere-2022-126, "Responses of CIPS/AIM Noctilucent Clouds to the Interplanetary Magnetic Field"**

The manuscript describes an analysis of space based observations of Noctilucent Clouds, also called Polar Mesospheric Clouds.
Observations between 2007 and 2017 are used and a correlation study with the IMF is performed on a day-to-day basis. The paper is well structured and reads in most parts well.

We thank the anonymous referee #1 for the valuable comments. The suggestions are very constructive and have been taken into accounts in the revised paper. In the following the remarks are responded point by point.

The analysis has a couple of major flaws that make the results questionable:

**Tides and observational effects:**

Tides at the cloud altitude are known to have a large effect on cloud occurrence and brightness, and other properties. Orbit changes and changes in the local time of the ascending and descending node might affect the correlation coefficients. A discussion is needed.

We are appreciated for this comment and agreed that the tide effects are very important in NLC variations. We have investigated and confirmed that the correlation coefficients will not be affected when the tide effects are taken into consideration. The relevant results have been discussed in the revised manuscript and listed here:

NLCs are dominantly influenced by the solar tides with the diurnal variation, and the NLCs occurrences are usually more frequent at the local time of morning (Fiedler & Baumgarten, 2018; Stevens et al., 2017). In addition, the NLCs can also be affected by the lunar tides, and the longitudinal variations in NLCs attributed to the non-migrating lunar tides have been found (Liu et al., 2016; von Savigny et al., 2017). To check whether the local time differences between the descending and ascending branches of the AIM satellite will affect the results, we separate the CIPS data of the descending and ascending branches into two groups. Similarly, in order to check the longitudinal variations, the CIPS data are divided into two groups in term of the longitude ranges of (-180°,0°) and (0°,180°). The correlation coefficients for the above two scenarios have been calculated and listed in Table 1, and the results for all of them are consistent with the results shown in Fig.2. In summary, the correlations coefficients are found not affected by the local time variations and longitudinal variations in the CIPS data caused by the tide effects, this further proves that our results are robust.

**Table 1.** The correlation coefficient of NLC properties with $IMF\ B_y$ under different data selections of satellite branches and longitudinal ranges.

| Data selections | $r_m$ (SH) | $r_m$ (NH) | $Alb_m$ (SH) | $Alb_m$ (NH) | $IWC_m$ (SH) | $IWC_m$ (NH) | $FO$ (SH) | $FO$ (NH) |
|---|---|---|---|---|---|---|---|---|
| All | 0.25±0.04 | -0.13±0.04 | 0.16±0.08 | -0.10±0.07 | 0.11±0.08 | -0.05±0.07 | 0.12±0.08 | -0.03±0.07 |
| Ascending | 0.23±0.04 | -0.09±0.04 | 0.14±0.07 | -0.07±0.06 | 0.10±0.07 | -0.05±0.06 | 0.09±0.07 | -0.00±0.07 |
| Descending | 0.19±0.06 | -0.15±0.06 | 0.15±0.08 | -0.10±0.07 | 0.09±0.08 | -0.04±0.07 | 0.13±0.09 | -0.05±0.06 |
| (-180°~0°) | 0.19±0.07 | -0.08±0.04 | 0.15±0.06 | -0.09±0.07 | 0.08±0.07 | -0.05±0.07 | 0.06±0.07 | -0.03±0.05 |
| (0°~180°) | 0.24±0.05 | -0.13±0.04 | 0.12±0.08 | -0.08±0.05 | 0.09±0.09 | -0.03±0.06 | 0.13±0.08 | -0.12±0.06 |

**Microphysics:**

The authors provide no detailed discussion about microphysical aspects that are well elaborated in literature (e.g., Rapp and Thomas, 2006 and references therein). Instead, they mention "coagulation", which is less relevant (unimportant) for mesospheric clouds. For example, IWC, brightness, and radius have a strong relation to each other. Since the detection threshold of CIPS depends on the particle size, it should be discussed how this affects the small particle size cutoff and its changes (e.g. Fig. 6).

> Coincidently, the Referee #2 was also very concerned about the microphysical mechanism. We have proposed a new microphysical mechanism, which emphasizes the role of the charged meteoric smoke particles (MSPs) and the nucleation process, as stated in the reply to Referee #2.
>
> As pointed out by the reviewer that the 'coagulation' is unimportant in NLCs, we decide to remove the relevant discussion.
>
> With regards to the relationship between NLC properties, we noticed that the detection threshold of CIPS for ice particles with 10-15 nm radii has been used to explain the opposite changes of the ice particle radius and ice particle concentration in NLCs during gravity waves (Gao et al., 2018). Meanwhile, simulations also confirm the opposite variations of ice particle radius and concentration from the view of the nucleation process in NLCs (Wilms et al., 2016). Both the above two explanations have been applied in the revised manuscript to discuss the relation between NLC properties.

**Electron densities:**

A discussion about the state of knowledge on IMF effects on the electron density at cloud altitudes is needed. E.g. in case IMF effects are longitude dependent, the results may be different for ascending and descending nodes. Since the electron density is relevant for particle charging in the dusty plasma environment, it is a key parameter.

> We are fully agreed with the reviewer that the electron density plays a key role in the link between $IMF$ $B_y$ and NLCs, especially for the charging process of MSPs. A new microphysical mechanism involving the electron density has been proposed. Please find the new mechanism in the Discussion part of the revised paper.
>
> The results of correlation coefficients for different branches as well as different longitudinal regions have been investigated in the previous responses to the tide effects. As shown in Table 1, the longitudinal effect of $IMF$ $B_y$ on ionospheric potential caused by the dipole tilt of geomagnetic field is insignificant or too small to be observed. In fact, studies usually concern more about the latitude variations of the ionospheric potential changes induced by the $IMF$ $B_y$, which are confirmed in our Fig. 4 and Fig. 5.

**PMSE**

A discussion of radar echoes associated with icy particles (PMSE) is completely missing. These radar echoes are caused/affected by electron density fluctuations and icy particles. Following the authors "$IMF$ $B_y$ - ionospheric potential - NLCs microphysics - NLCs brightness'", they are likely more directly affected than NLCs.

> We thank for the useful comment. The PMSE are well known to be closely related with the charged ice particles in NLCs. In the revised manuscript the PMSE have been discussed as follows:

"Polar mesosphere summer echoes (PMSE) are very strong radar echoes scattered by the electron number density irregularities at the polar summer mesopause altitudes of about 75-100 km, and the electron structures are thought to be caused by the neutral air turbulence in combination with the charged ice particles in the NLCs (Rapp and Lübken, 2004). Note that the NLCs are absent in the winter hemisphere, whereas polar mesosphere winter echoes (PMWE) were still observed at much lower altitudes of 55- 85 km. PMWE are suggested to be caused by the neutral air turbulence together with the charged MSPs (Strelnikov et al., 2021). A possible link is expected to exist between PMSE/PMWE with the $IMF$ $B_y$ for two reasons: First, the PMSE is sensitive to ice particle radius and concentration, due to the ice particle can affect the diffusion of electrons (Rapp and Lübken, 2004). Our results show that the ice particle radius is sensitive to solar wind, thus it is necessary to check whether this response has further influence on the PMSE. Second, as mentioned in the above microphysical process, the $IMF$ $B_y$ is supposed to have a major effect on the charging process of MSPs, and the latter play a more direct role in PMSE/PMWE. In conclusion, to investigate the response of PMSE/PMWE to $IMF$ $B_y$ will be helpful for understanding the link between solar wind and mesosphere, while the relevant work is beyond the scope of this paper."

**Specific comments:**

Line 106: Due to the large number of noisy lines in Figure 1, a correlation is not visible. A more convincing display would help.

Agreed. A new figure for the 2008/2009 summer season in SH has been plotted to make the correlation more visible.

Line 112: Figure 2 does not provide uncertainties. How significant are the year-to-year changes shown?

Done. The uncertainties have been shown by adding the error bar for the standard deviation of the mean in Fig. 2.

Line 126: It may be more convincing if negative lag days are also shown in Fig. 3.

Agreed. The results for negative lag days have been added in Fig. 3.

Line 127: *"In previous studies of the link between Ly-α and NLCs, the proposed mechanism involving photodissociation, heating, or circulation all required longer time"*: What causes the *"longer time"*, for example, for photodissociation? A more detailed discussion/references may help.

Done. We have cited the reference of Shapiro et al. (2012) to describe the time lag of photodissociation, the references of Thomas et al. (2015) and Thurairajah et al. (2017) to present the observed lag time for the responses of NLCs to the *Ly-α* solar irradiance.

**References:**

Fiedler, J. and Baumgarten, G.: Solar and lunar tides in noctilucent clouds as determined by ground-based lidar, Atmospheric Chemistry and Physics, 18, 16051–16061, https://doi.org/10.5194/acp-18-16051-2018, 2018.

Gao, H., Li, L., Bu, L., Zhang, Q., Tang, Y., and Wang, Z.: Effect of small-scale gravity waves on

polar mesospheric clouds observed from CIPS/AIM, Journal of Geophysical Research: Space Physics, 123, 4026–4045, https://doi.org/10.1029/2017JA024855, 2018.

Liu, X., Yue, J., Xu, J., Yuan, W., Russell III, J. M., Hervig, M. E., and Nakamura, T.: Persistent longitudinal variations in 8 years of CIPS/AIM polar mesospheric clouds, Journal of Geophysical Research: Atmospheres, 121, 8390–8409, https://doi.org/10.1002/2015JD024624, 2016.

Rapp, M. and Lübken, F.-J.: Polar mesosphere summer echoes (PMSE): Review of observations and current understanding, Atmospheric Chemistry and Physics, 4, 2601–2633, https://doi.org/10.5194/acp-4-2601-2004, 2004.

Shapiro, A. V., Rozanov, E., Shapiro, A. I., Wang, S., Egorova, T., Schmutz, W., and Peter, Th.: Signature of the 27-day solar rotation cycle in mesospheric OH and H2O observed by the Aura Microwave Limb Sounder, Atmospheric Chemistry and Physics, 12, 3181–3188, https://doi.org/10.5194/acp-12-3181-2012, 2012.

Stevens, M. H., Liebermann, R. S., Siskind, D. E., McCormack, J. P., Hervig, M. E., and Englert, C. R.: Periodicities of polar mesospheric clouds inferred from a meteorological analysis and forecast system, Journal of Geophysical Research: Atmospheres, 122, 4508–4527, https://doi.org/10.1002/2016JD025349, 2017.

Strelnikov, B., Staszak, T., Latteck, R., Renkwitz, T., Strelnikova, I., Lübken, F.-J., Baumgarten, G., Fiedler, J., Chau, J. L., Stude, J., Rapp, M., Friedrich, M., Gumbel, J., Hedin, J., Belova, E., Hörschgen-Eggers, M., Giono, G., Hörner, I., Löhle, S., Eberhart, M., and Fasoulas, S.: Sounding rocket project "PMWE" for investigation of polar mesosphere winter echoes, Journal of Atmospheric and Solar-Terrestrial Physics, 218, 105596, https://doi.org/10.1016/j.jastp.2021.105596, 2021.

Thomas, G. E., Thurairajah, B., Hervig, M. E., von Savigny, C., and Snow, M.: Solar-induced 27-day variations of mesospheric temperature and water vapor from the AIM SOFIE experiment: Drivers of polar mesospheric cloud variability, Journal of Atmospheric and Solar-Terrestrial Physics, 134, 56–68, https://doi.org/10.1016/j.jastp.2015.09.015, 2015.

Thurairajah, B., Thomas, G. E., von Savigny, C., Snow, M., Hervig, M. E., Bailey, S. M., and Randall, C. E.: Solar-induced 27-day variations of polar mesospheric clouds from the AIM SOFIE and CIPS experiments, Journal of Atmospheric and Solar-Terrestrial Physics, 162, 122–135, https://doi.org/10.1016/j.jastp.2016.09.008, 2017.

von Savigny, C., DeLand, M. T., and Schwartz, M. J.: First identification of lunar tides in satellite observations of noctilucent clouds, Journal of Atmospheric and Solar-Terrestrial Physics, 162, 116–121, https://doi.org/10.1016/j.jastp.2016.07.002, 2017.

Wilms, H., Rapp, M., and Kirsch, A.: Nucleation of mesospheric cloud particles: Sensitivities and limits, Journal of Geophysical Research: Space Physics, 121, 2621-2644, https://doi.org/10.1002/2015JA021764, 2016.